# Mitochondrial Peptide Humanin Facilitates Chemoresistance in Glioblastoma Cells

**DOI:** 10.3390/cancers15164061

**Published:** 2023-08-11

**Authors:** Jorge A. Peña Agudelo, Matías L. Pidre, Matias Garcia Fallit, Melanie Pérez Küper, Camila Zuccato, Alejandro J. Nicola Candia, Abril Marchesini, Mariana B. Vera, Emilio De Simone, Carla Giampaoli, Leslie C. Amorós Morales, Nazareno Gonzalez, Víctor Romanowski, Guillermo A. Videla-Richardson, Adriana Seilicovich, Marianela Candolfi

**Affiliations:** 1Instituto de Investigaciones Biomédicas (INBIOMED, UBA-CONICET), Facultad de Medicina, Universidad de Buenos Aires, Consejo Nacional de Investigaciones Científicas y Técnicas, Buenos Aires C1121A6B, Argentina; jarmando2192@gmail.com (J.A.P.A.); mati.garciafallit@gmail.com (M.G.F.); melperezkuper@gmail.com (M.P.K.); camilazuccato@gmail.com (C.Z.); alenicola90@gmail.com (A.J.N.C.); gonzalez.nazareno1@gmail.com (N.G.); adyseili@fmed.uba.ar (A.S.); 2Instituto de Biotecnología y Biología Molecular (IBBM, UNLP-CONICET), Facultad de Ciencias Exactas, Universidad Nacional de La Plata, La Plata B1900, Argentina; mlpidre@biol.unlp.edu.ar (M.L.P.); abrilmarchesini@biol.unlp.edu.ar (A.M.); amorosleslie@biol.unlp.edu.ar (L.C.A.M.); victor@biol.unlp.edu.ar (V.R.); 3Departamento de Química Biológica, Facultad de Ciencias Exactas y Naturales, Universidad de Buenos Aires, Buenos Aires C1428BFA, Argentina; 4Fundación Para la Lucha Contra las Enfermedades Neurológicas de la Infancia (FLENI), Buenos Aires C1121A6B, Argentina; mvera.ext@fleni.org.ar (M.B.V.); gvidela@fleni.org.ar (G.A.V.-R.); 5Cátedra de Fisiología Animal, Facultad de Ciencias Veterinarias, Universidad de Buenos Aires, Buenos Aires C1428BFA, Argentina; edesimone@fvet.uba.ar (E.D.S.); cgiampaoli@fvet.uba.ar (C.G.); 6Departamento de Biología Celular e Histología, Facultad de Medicina, Universidad de Buenos Aires, Buenos Aires C1121A6B, Argentina

**Keywords:** glioblastoma, humanin, FPR2, chemotherapy

## Abstract

**Simple Summary:**

Glioblastoma (GBM) is an aggressive brain tumor with high resistance to chemotherapy. Understanding the underlying molecular mechanisms of its chemoresistance is essential for providing effective therapeutic strategies. Although humanin (HN) analogues have been proposed for the treatment of chronic diseases, in this study we show that these analogues contribute to chemoresistance in GBM cells. Here, we present evidence that HN, a mitochondrial peptide with cytoprotective properties, and its membrane receptor FPR2 are expressed in GBM cells and upregulated by chemotherapy. We found that FPR2 mediates the cytoprotective effects of HN in glioma cells. Thus, we inhibited the expression of HN using gene therapy vectors, improving the sensitivity of GBM cells to chemotherapy. These findings suggest that HN and its receptor FPR2 could be involved in the progression of GBM and they may represent promising therapeutic targets to improve the efficacy of chemotherapy in these patients.

**Abstract:**

Humanin (HN) is a mitochondrial-derived peptide with robust cytoprotective effects in many cell types. Although the administration of HN analogs has been proposed to treat degenerative diseases, its role in the pathogenesis of cancer is poorly understood. Here, we evaluated whether HN affects the chemosensitivity of glioblastoma (GBM) cells. We found that chemotherapy upregulated HN expression in GBM cell lines and primary cultures derived from GBM biopsies. An HN analog (HNGF6A) boosted chemoresistance, increased the migration of GBM cells and improved their capacity to induce endothelial cell migration and proliferation. Chemotherapy also upregulated FPR2 expression, an HN membrane-bound receptor, and the HNGF6A cytoprotective effects were inhibited by an FPR2 receptor antagonist (WRW4). These effects were observed in glioma cells with heterogeneous genetic backgrounds, i.e., glioma cells with wild-type (wtIDH) and mutated (mIDH) isocitrate dehydrogenase. HN silencing using a baculoviral vector that encodes for a specific shRNA for HN (BV.shHN) reduced chemoresistance, and impaired the migration and proangiogenic capacity of GBM cells. Taken together, our findings suggest that HN boosts the hallmark characteristics of GBM, i.e., chemoresistance, migration and endothelial cell proliferation. Thus, strategies that inhibit the HN/FPR2 pathway may improve the response of GBM to standard therapy

## 1. Introduction

Glioblastoma (GBM) is the most aggressive and frequent primary malignant brain tumor in adults. The standard treatment for patients with GBM, which includes surgical removal, radiotherapy and chemotherapy, has not changed in almost 20 years and their prognosis is dismal [1]. GBM is characterized by its invasiveness and intrinsic resistance to conventional therapy. Thus, the tumor recurs and kills the patients, whose median survival is 12–15 months [2,3]. For these reasons, it is necessary to find therapeutic targets that improve the response of GBM cells to treatment.

Humanin (HN) is a mitochondrial peptide with a potent cytoprotective effect in many cell types. HN can interact with proteins of the Bcl-2 family and inhibit the intrinsic apoptotic pathway [4,5,6,7]. HN can also be released and it has been described to interact with two membrane receptors: a trimetric receptor, composed of ciliary neurotrophic factor receptor (CNTFR), IL27R (WSX-1) and the 130 kDa glycoprotein (gp130) that can trigger the activation of RAS/MAPK, PI3K, JNK and STAT3; or the FPR2 receptor that induces the activation of the extracellular signal-regulated kinase (ERK) pathway [6,8]. Cells can also take up exogenous HN, which rapidly localizes in the mitochondria [9,10]. A robust antiapoptotic effect of HN has been observed in several cell types, i.e., pancreatic β cells, germ cells, neurons, endothelial cells and secretory cells of the anterior pituitary [11,12]. Thus, the administration of HN has been proposed to treat chronic medical conditions, such as diabetes, and neurodegenerative and cardiovascular diseases (US patents 8653027 B2, WO2008153788 A2, US20130123168 A1).

Since overexpression of HN was detected in gastric [13], bladder [14], pituitary [15] and breast cancer cells [16], we and others have proposed that upregulation of HN could play a role in tumorigenesis [12]. Although HN has been shown to protect normal cells from chemotherapy [11,12], the effect of HN on tumor pathogenesis is poorly understood and controversial [8]. While some authors have proposed that certain HN analogs could improve the response of tumor cells to chemotherapy, the expression of HN in cancer biopsies has been associated with the development of chemoresistance [13]. Our previous results indicate that HN exerts a strong cytoprotective effect in breast cancer cells, facilitating tumor progression and chemoresistance in experimental breast cancer models [16]. The aim of our study was to assess whether HN affects the hallmark features that characterize GBM: i.e., chemoresistance, migration and angiogenesis. We evaluated the effect of exogenous and endogenous HN on murine and human GBM cells. Our findings suggest that HN elicits a strong cytoprotective effect in GBM cells via the FPR2 membrane-bound receptor, facilitating tumor cell migration, angiogenesis and chemoresistance in glioma cells of heterogeneous genetic backgrounds. Thus, the inhibition of the HN/FPR2 axis could improve the response of GBM to standard therapy.

## 2. Materials and Methods

### 2.1. Drugs

HNGF6A, a humanin analog substituted with serine and glycine at the 6th and 14th amino acids, respectively; and the selective antagonist of formyl peptide receptor 2 (FPR2), WRW4, were obtained from Tocris Biosciences, with catalog numbers #5154 and #2262, respectively. HNGF6A and WRW4 were dissolved separately in 1 mL of water free of DNAses and RNAses to obtain a concentrated solution of 1 mg/mL. The solution was gently mixed until the peptides were completely dissolved. The final concentration of the solution was 387 µM and 905 µm, for the solutions free of DNAses and RNAses, respectively. Subsequently, aliquots of the dissolved peptide were made in fractions of 25 µL to facilitate storage and avoid freeze–thaw cycles that could affect the stability of the peptide. The aliquots were kept at −20 °C until their later use in experiments. The culture media, Dulbecco’s Modified Eagle Medium (DMEM; Cat# 12100046), Dulbecco’s Modified Eagle Medium: F-12 Nutrient Mix (DMEM/F-12; Cat# 12500062), Neurobasal Medium (Cat# 21103049), B-27 and N-2 supplements (Cat# A35828-01 and Cat# 17502-048, respectively), Geltrex LDEV-Free Reduced Growth Factor Basement Membrane Matrix (Cat# A14132-02), penicillin–streptomycin (Cat# 15140122), trypsin–EDTA (0.025%, Cat# 25200114) and Lipofectamine 2000 (Cat# 11668019) were obtained from Gibco (Invitrogen, Carlsbad, CA, USA). Fetal bovine serum (FBS) was acquired from Natocor (Cordoba, Argentina). Cisplatin (CIS) was obtained from Microsules (Buenos Aires, Argentina) and temozolomide (TMZ) was obtained from Sigma (St. Louis, MO, USA), and the materials are indicated below.

### 2.2. Cell Culture

GL26 and U251-MG GBM cell lines and the neurospheres derived from murine (wtIDH and mIDH) gliomas were kindly donated by Dr Maria G Castro (University of Michigan School of Medicine, Ann Arbor, MI, USA) [17]. Cells were cultured in DMEM containing 5% FBS and 1% penicillin–streptomycin. To collect the cells, 0.05% trypsin–EDTA was used, and then they were counted with trypan blue.

Murine neurospheres were cultured in DMEM-F12 supplemented with 1% penicillin–streptomycin, 1X B-27, 1X N-2, 100 µg/mL Normocin, 20 ng/mL bFGF and 20 ng/mL EGF. The neurospheres were collected and disaggregated using accutase and then counted with trypan blue.

Patient-derived G01 and G09 cells were obtained from mutated IDH (mIDH) and wild-type IDH (wtIDH) glioma biopsies, respectively. The use of these cultures for biomedical research was approved by the Ethics Committee in Biomedical Research of the Foundation for the Fight Against Neurological Diseases in Childhood (FLENI, Buenos Aires, Argentina). These cells were cultured on Geltrex-coated Petri dishes with serum-free neurobasal medium supplemented with glucose, sodium pyruvate, PBS-BSA 7.5 mg/mL, 1X B27, 1X N2, 20 ng/mL bFGF and EGF, 2 mM L-glutamine, 2 mM non-essential amino acids, and 50 U/mL penicillin/streptomycin. The cells were harvested with accutase and counted with trypan blue.

### 2.3. Immunofluorescence

HN and FPR2 expression was evaluated by fluorescent microscopy, as previously described [16]. Briefly, 70,000 GBM cells were seeded on coverslips in 24-well plates and incubated with or without cisplatin (2 µM) for 48 h. Then, cells were fixed with 4% PFA and subjected to antigen retrieval with citrate buffer (pH 6) in a microwave at 350 W for 5 min. Cells were permeabilized with a solution of TBS–0.5% Triton–0.1% sodium azide. To block non-specific binding sites, a solution of TBS–0.2% Triton–0.1% sodium azide and 10% goat serum was used for 1 h. Subsequently, cells were incubated with an anti-HN (1:100, Novus Biol cat# NB100-56877SS) or anti-FPR2 (1:100, Novus Biol cat# NLS1878SS) antibody in 0.2% TBS–Triton–0.1% sodium azide and 1% goat serum overnight. The next day, cells were washed and incubated with an anti-rabbit IgG antibody (1:200, Vector Laboratories Inc., Newark, CA, USA) to HN and Alexa Fluor 488 conjugated anti-rabbit antibody (1:200, Invitrogen cat# A-11070) to FPR2. Finally, cells were stained with DAPI (4′,6-diamidino-2-phenylindole) used at a concentration of 5 µg/mL and mounted on slides with Vectashield (Vector Laboratories, Inc., Newark, CA, USA). Visualization of cells was performed using a fluorescent light microscope (Axiophot; Carl Zeiss, Jena, Germany). Negative controls were created by incubating cells without the primary antibody.

### 2.4. HN and FPR2 Detection by Flow Cytometry

Cells were harvested using 0.025% EDTA–trypsin and washed with PBS. Subsequently, they were fixed with 2% PFA and permeabilized with 0.1% saponin (MP Biomedicals, Inc., Solon, OH, USA) for 10 min. Cells were then incubated with anti-HN antibody (1:100, Novus Biol cat# NB300-246) or anti-FPR2 antibody (1:100, Novus Biol cat# NLS1878SS) in PBS for 1 h at room temperature. Finally, cells were incubated with Alexa Fluor 488 conjugated anti-rabbit antibody (1:100, Invitrogen cat# A-11070) in PBS for 1 h at room temperature. After washing, cells were resuspended in PBS and analyzed by flow cytometry using a FACScalibur instrument (Becton Dickinson, Franklin Lakes, NJ, USA). Data obtained from flow cytometry analysis were processed using FlowJo v10 software, as previously described [16,18].

### 2.5. RNA Isolation, RT-PCR and qRT-PCR

For the extraction of total RNA from U251-MG cells treated with chemotherapy, Trizol (Thermo Scientific, Rockford, IL, USA) was used following the manufacturer’s instructions. The cDNA synthesis was performed using MMLV reverse transcriptase (Promega, Madison, WI, USA). Quantitative RT-PCR assays were conducted using SYBR Green-ER qPCR SuperMix Universal (Thermo Scientific, Rockford, IL, USA). Primers used were as follows: hHN: forward 5′-TGTCAACCCAACACAGGCATG-3′; reverse 5′-AAACAGGCGGGGTAAGATTTG-3′; as the internal control, RPL7: forward 5′-AATGGCGAGGATGGCAAG-3′, reverse 5′-TGACGAAGGCGAAGAAGC-3′. PCR amplification was carried out using a StepOnePlus real-time PCR system (Applied Biosystems, Foster City, CA, USA).

### 2.6. Propidium Iodide Exclusion Assay

A total of 60,000 cells were seeded per well in 24-well plates and treated according to the established conditions and incubation times. For sample preparation, independent tubes were used, and supernatants and cells previously detached with 0.025% trypsin–EDTA were collected. The samples were centrifuged for 5 min at 1500 rpm and the supernatant was discarded. For the preparation of the propidium iodide (PI) stock solution, 1 mg of PI was dissolved in 1 mL of distilled water; then, the working solution was prepared using 1 µL of the stock solution in 100 µL of PBS. The cells were resuspended with 200 µL of the working solution and immediately analyzed by flow cytometry. Dead cells were identified by emitting fluorescence upon excitation at 488 nm.

### 2.7. BrdU Cell Proliferation Assay

Cell proliferation was evaluated by incorporation of bromodeoxyuridine/5-bromo-2′-deoxyuridine (BrdU; Sigma Aldrich, Roche #Cat. 11647229001, St. Louis, MO, USA). Absorbance was determined using a 96-well plate spectrophotometer (Bio-Rad, Model 550, Hercules, CA, USA) at 490 nm, as previously described [16,19,20].

### 2.8. Cell Viability

Cell viability was evaluated using the 3-(4,5-dimethylthiazol-2-yl)-2,5-diphenyltetrazolium bromide (MTT) assay (Molecular Probes, Invitrogen, Thermo Fisher Scientific, Waltham, MA, USA), as previously described [16,19,20].

### 2.9. Clonogenic Assay

Initially, 5000 cells were seeded in a 96-well plate according to the experimental conditions and established incubation times. Cells were then harvested with 0.025% trypsin–EDTA and counted with trypan blue to seed a density of 2000 cells per well in 6-well plates. After 10 days, cells were fixed in methanol for 10 min, washed with PBS, and stained with Giemsa. The number of colonies containing a minimum of 50 cells (colony-forming unit, CFU) were counted using a binocular stereomicroscope and the clonogenic fraction was calculated based on the number of cells seeded relative to the number of clones formed.

### 2.10. Migration Assay

A wound healing assay was performed to assess the migratory activity of EA.hy926 endothelial cells, as well as U251-MG human GBM cells incubated with or without HNGF6A directly or with the conditioned medium of HNGF6A-treated cells. Cells were grown to confluence and incubated in DMEM with 1% FBS containing HNGF6A (1.25 µM) or with conditioned medium from cells treated with HNGF6A (1.25 µM) for 48 h. After 24 h, a wound was made with the tip of a micropipette, the medium was removed, and two washes with PBS were performed to remove cell debris. Afterwards, fresh media containing the peptide or the conditioned media was added. Cell migration into the free space was photographed and measured using ImageJ software (Version: 1.53k) at several time points.

### 2.11. Zymography

U251-MG human GBM cells were incubated in the presence of 1.25 µM solution for 48 h. Conditioned medium was collected and MMP gelatinolytic activity was assessed by zymography as previously described [20].

### 2.12. Plasmid Construction and Transfections

The shRNA comprising the hHN RNA sequence (shRNA hHN: CCCGTGAAGAGGCGGGCATAAAAGTTCTCTTTATGCCCGCCTCTTCACGGGTTTTTT) was synthesized and fused with the U6 promoter and cloned into a bicistronic pUC57 vector (p. shHN). To detect transfected cells, the construct contains the coding sequence for the fluorescent reporter protein dTomato under the control of the immediate early (IE) promoter of the cytomegalovirus (CMV). GBM cells were transfected with 1 μg of p. shHN or control plasmid DNA using Lipofectamine 2000 and incubated for 48 h. To evaluate transfection efficiency, cells were washed with PBS, fixed with 4% PFA for 10 min, stained with DAPI, and mounted with Vectabom for subsequent visualization through fluorescence microscopy. To evaluate the efficiency of transfection or to evaluate HN expression, cells were incubated with cisplatin (2 µM) for 72 h, and cell viability was measured using MTT.

### 2.13. Generation of Recombinant Baculoviruses

We developed a recombinant baculoviral vector, AcMNPV, encoding a specific shRNA targeting HN (BV-shHN) to silence its expression. To achieve this, we cloned the shRNA cassette into the pBacPAK9 transfer vector digested with EcoRV-NotI (Clontech, Mountainview, CA, USA). The AcMNPV sequences present in the vector allowed for homologous recombination with the viral DNA in insect cells to transfer the expression cassette to the viral polyhedrin locus. To generate the recombinant baculovirus, we co-transfected the recombinant pBacPAK9 into the Trichoplusia ni BTI-TN-5B1-4 insect cell line (High FiveTM; Thermo Fisher Scientific, Waltham, MA, USA) with bApGOZA DNA. Following infection, cells were maintained in Grace medium (Thermo Fisher Scientific) supplemented with 10% FBS at 27 °C until signs of infection were detected. The same strategy was used to generate the control baculovirus (BV-Control) that expresses only the green fluorescent protein citrine without the shRNA sequence. Recombinant citrine expression was verified by fluorescence microscopy. The BVs were titrated on a High Five^TM^ insect cell monolayer as PFU, and these titers coincided with the infection foci of the citrine reporter gene.

### 2.14. Baculovirus-Mediated Gene Transduction

Murine GBM cells (GL26) were incubated with BV.Control or BV.shHN (750 pfu/cell) for 2 h in DMEM, and then supplemented medium was added. After 48 h, post transduction viability and migration assays were performed.

### 2.15. Meta-Analysis of HN and FPR2 Expression

Clinical, genomic and transcriptomic data from GBM patients were obtained from the public datasets of TCGA Pan-Cancer. Transcriptomic data from normal brain samples were obtained from the GTEx dataset. Clinical information and mRNA expression data of HN (MT-RNR2) and FPR2 were downloaded from https://tcga-data.nci.nih.gov/via (accessed on 4 August 2023) the Xena Browser developed by UCSC.

### 2.16. Statistical Analysis

Data were analyzed using GraphPad Prism software, version 8 (GraphPad Software). Differences in BrdU incorporation, cell death, clonogenic ratio and MTT data were analyzed by analysis of variance (ANOVA) followed by Tukey’s post-hoc test. Differences in HN expression levels assessed by qPCR, flow cytometry, as well as endothelial proliferation and zymographic activity were analyzed using Student’s *t*-test. Nonlinear correlation analysis was used to analyze differences in the scratch assay. Kaplan–Meier curves were employed to estimate the progression-free interval (PFI) and overall survival (OS) between groups stratified according to HN/FPR2 expression levels in HN/FPR2low and HN/FPR2high, using median expression values as the cut-off. The log rank test was used to analyze differences between the survival curves. Differences between groups were considered significant when *p* < 0.05. All experiments were performed at least twice.

## 3. Results

### 3.1. Chemotherapy Upregulates HN Expression in GBM Cells

We first assessed HN expression by immunofluorescence in the human GBM cell line U251-MG, as well as in cell cultures derived from a GBM biopsy (wtIDH glioma, G09) [21] incubated with or without cisplatin. HN expression was detected in all cells studied, but it was visibly increased when they were treated with chemotherapy (Figure 1A). Furthermore, the intracellular distribution of HN seemed to change in response to chemotherapy. We also assessed the effect of cisplatin on HN expression by flow cytometry in U251-MG human GBM cells. We observed that cisplatin induced a significant increase in HN protein levels (Figure 1B). However, when we explored the expression of HN at the transcriptional level by qPCR (Figure 1C and Appendix A), we did not find significant differences. These findings suggest the possibility that the translation of HN mRNA to protein had already occurred before the incubation time analyzed, when striking changes were detected in HN protein levels.

### 3.2. Exogenous HN Worsens the Hallmark Characteristic of GBM Cells

#### 3.2.1. Chemoresistance

The administration of exogenous HN has been proposed as a therapeutic strategy for the treatment of various chronic and neurodegenerative conditions [8]. Considering that we have previously reported that HN exerts a potent cytoprotective effect in pituitary tumor cells [15,18] and in HER2+ and triple-negative breast cancer cells [16], we assessed whether exogenous HN can also modulate the response of GBM cells to chemotherapy. First, we evaluated the effect of an HN analogue peptide (HNGF6A) at different doses on the response of U251-MG human GBM cells to cisplatin, as assessed by the MTT assay (Figure 2A). We observed a concentration-dependent cytoprotective effect of the HN analog that inhibited cisplatin-induced cytotoxicity in U251-MG cells. Since 1.25 µM HNGF6A was the lowest concentration that fully inhibited the cytotoxic effect of cisplatin, the following experiments were performed using that concentration. We next evaluated the effect of HNGF6A on the chemoresistance of U251-MG cells. We evaluated viability by MTT assay (Figure 2B), proliferation by BrdU incorporation (Figure 2C), cell death by PI exclusion (Figure 2D) and clonogenic capacity (Figure 2E) in U251-MG cells treated with HNGF6A and cisplatin. We found that HNGF6A inhibited the cytotoxic, antiproliferative and proapoptotic effects of cisplatin in these cells. Interestingly, we found that HNGF6A increased the viability of GBM cells per se (Figure 2B). While GBM cells incubated with cisplatin completely lost their ability to form clones, incubation with HNGF6A partially restored the clonogenic capacity of these cells (Figure 2E). In order to evaluate whether HN could also affect the cytotoxic response of lower-grade glioma cells, we used neurospheres derived from biopsies of astrocytoma patients harboring the IDH mutation (mIDH) [21]. We also used neurospheres derived from mIDH gliomas genetically engineered in mice, which harbor additional hallmark molecular features of mIDH astrocytomas, i.e., mutant p53 and ATRX loss [17]. It is important to highlight that, in addition to GBM, tumors considered to be of lower grade also have an unfavorable prognosis, which justifies the need to investigate their distinctive characteristics. We found that HNGF6A also impaired the cytotoxic effect of cisplatin in murine and human mIDH glioma neurospheres (Figure 2F).

#### 3.2.2. Tumor Cell Migration

Considering that invasion is a hallmark feature of GBM, we evaluated whether HN affects the migration of GBM cells. We performed a wound assay in GBM U251-MG cells incubated in the presence of HNGF6A or with conditioned medium from GBM U251-MG cells previously exposed to HNGF6A (Figure 3). When GBM cells were incubated directly with the HN analog, there were no significant differences in wound closure (Figure 3A). However, migration was increased in cells incubated with conditioned medium from HNGF6A-treated cells (Figure 3A). Next, we evaluated the effect of HNGF6A on the secretion of active metalloproteases (MMP) MMP-2 and MMP-9, which are essential for tumor cell invasion, by means of zymography in media from GBM U251-MG cells treated with HNGF6A. The addition of exogenous HN did not significantly change the secretion of active MMPs (Figure 3B and Appendix A).

#### 3.2.3. Endothelial Cell Migration and Proliferation

Since angiogenesis and endothelial proliferation are also hallmark features of GBM, we evaluated the effect of HNGF6A in EA.hy926 human endothelial cells (Figure 3C,D). We assessed the migration of EA.hy926 endothelial cells by the wound closure assay and their proliferation by BrdU incorporation after incubation with HNGF6A or with conditioned media from HNGF6A-treated GBM cells. While HNGF6A accelerated the migration and boosted the proliferation of EA.hy926 endothelial cells, conditioned media from GBM cells treated with this peptide did not affect these characteristics in endothelial cells (Figure 3D).

### 3.3. FPR2 Mediates the Cytoprotective Effects of HN in GBM Cells

Taking into account that endogenous and exogenous HN can interact with membrane receptors on target cells, we next assessed FPR2 expression in U251-MG human GBM cells. Using immunofluorescence (Figure 4A), we observed the basal expression of FPR2 in these cells, which was upregulated in response to cisplatin. Cisplatin-induced upregulation of FPR2 was quantified and confirmed by flow cytometry (Figure 4B). The upregulation of HN and FPR2 suggests that they could be involved in the chemoresistance of GBM cells. In this sense, to evaluate whether FPR2 was involved in the cytoprotective effect of HN in GBM cells, we evaluated the effect of HNGF6A in human GBM U251-MG cells as well as in mIDH and wtIDH murine glioma neurospheres treated with cisplatin in the presence of a peptidic FPR2 receptor antagonist, WRW4 [22,23]. We observed that, in the presence of WRW4, HNGF6A was unable to inhibit the cytotoxic effect of cisplatin in human GBM cells (Figure 4C), as well as in wtIDH and mIDH glioma neurospheres (Figure 4E). Furthermore, WRW4 reduced the viability of GBM cells per se (Figure 4C). We also observed that FPR2 blockade restored the antiproliferative effect of cisplatin in GBM cells incubated in the presence of the HNGF6A (Figure 4D). These results suggest that FPR2 plays a crucial role in mediating the cytoprotective effects of HN in GBM cells.

### 3.4. Blockade of Endogenous HN Ameliorates the Hallmark Features of GBM Cells

#### 3.4.1. Chemoresistance

In view of our findings, we hypothesized that silencing endogenous HN could improve the sensitivity of GBM cells to chemotherapy. Thus, we developed a baculoviral (BV) vector encoding an shRNA specific for murine HN (BV.shHN) to silence its expression in GBM cells [16]. The construct encodes for the shHN under the control of the U6 promoter and for a reporter gene of the green fluorescent protein citrine under the control of the CMV promoter, which allows the evaluation of transduction efficiency by fluorescent microscopy [24]. As a control, we used a BV that only expresses citrine (BV.citrine) [24]. Tallying our previous results using BVs [24], both viral vectors efficiently transduced murine GBM cells (GL26) (Figure 5A). Since GL26 murine GBM cells also upregulated HN expression in response to cisplatin (Appendix A), we aimed to evaluate whether BV-mediated HN silencing affected the response of these cells to chemotherapy. We found that BV.shHN HN reduced the viability of GL26 murine GBM cells per se and sensitized them to the cytotoxic effect of cisplatin (Figure 5B).

To silence the expression of HN in human GBM cells we used a plasmid (p.shHN) that encodes for an shRNA specific for human HN under the control of the U6 promoter and the reporter gene for the red fluorescent protein dTomato (Figure 6A). Transfection of U251-MG cells with this plasmid inhibited their viability and improved the cytotoxic effect of cisplatin in U251-MG cells and cells derived from mIDH-glioma biopsies (G01) (Figure 6B).

Even though temozolomide (TMZ) has been used for almost 20 years as the standard of care for GBM patients, these cells are highly resistant to TMZ-mediated cytotoxicity. We found that treatment with TMZ upregulated HN expression in U251-MG cells (Figure 6C). When we evaluated whether HN silencing affects the sensitivity of GBM cells to TMZ, we found that p.shHN also improved the cytotoxic effect of TMZ in these cells (Figure 6C).

#### 3.4.2. Migration

We evaluated the effect of HN silencing on the migratory capacity of murine GL26 GBM cells using BV.shHN. Cells were transduced with BV.shHN, and 24 h later, the wound assay was performed. We observed that BV.shHN delayed the migration of these cells (Figure 5D), suggesting that endogenous HN facilitates GBM cell migration.

#### 3.4.3. Endothelial Cell Migration

We assessed the role of endogenous HN in promoting endothelial cell migration using conditioned medium from GL26 GBM cells transduced with BV.shHN. We found that the conditioned medium from BV.shHN-treated cells inhibited the migration of EA.hy926 endothelial cells (Figure 5C), suggesting that HN facilitates the secretion of proangiogenic factors from GBM cells.

### 3.5. HN and FPR2 Expression in GBM Biopsies

Considering that HN and FPR2 could promote the hallmark features of GBM, we aimed to evaluate their expression levels in GBM biopsies and normal brain tissue. Thus, we performed a meta-analysis using transcriptomic data deposited at the TCGA GBM and GTEx. We found that HN mRNA expression was higher in normal brain tissue that in GBM biopsies (Figure 7A). However, FPR2 expression levels were significantly upregulated in GBM biopsies compared to normal brain tissue (Figure 7B). To evaluate the potential prognostic role of these markers in glioma, we stratified GBM patients according to the expression levels of HN or FPR2. While local HN expression levels did not show any correlation with GBM patient progression or survival (Figure 7C), tumor FPR2 expression was associated with a worse prognosis, since patients with higher levels of FPR2 exhibited a reduced progression-free interval (PFI) and lower overall survival (OS) (Figure 7D).

## 4. Discussion

Since the phase III studies conducted by Stupp et al. [25] that led to the introduction of TMZ to treat newly diagnosed GBM patients, no significant improvements have arisen for these patients, whose survival is 12–15 months after diagnosis worldwide [3]. The treatment of GBM faces complex challenges due to the characteristics of these tumors, such as resistance to conventional therapies, frequent recurrence related to its infiltrative nature [26], aberrant neovascularization, its genetic and molecular heterogeneity [27,28], and the location of this tumor within the brain. A full understanding of the mechanisms that contribute to this resistance is essential for the development of strategies aimed at improving the response of these tumor cells to treatment. Although HN has been shown to be cytoprotective in normal cells exposed to chemotherapeutic drugs [12,29], the role of this peptide in the chemosensitivity of GBM cells has not been explored. In this study, we found the expression of HN and its membrane receptor FPR2 in GBM cells, which were further upregulated by chemotherapy. Interestingly, we noted a distinctive change in the intracellular distribution of HN in response to chemotherapy, suggesting that intracellular trafficking of this peptide could be involved in the resistance of GBM cells to chemotherapy. In fact, silencing of HN expression reduced the viability and chemoresistance of human and murine glioma cells with heterogeneous genetic backgrounds. Characterization of the mutational status of isocitrate dehydrogenase (IDH) 1 and 2 has become an essential component for the classification and prediction of the prognosis of diffuse gliomas in adults [30]. Tumors with a mutation in IDH (mIDH) have been shown to have a more favorable prognosis compared with GBM (wtIDH) [31,32]. Nevertheless, mIDH glioma patients also need therapeutic alternatives, as their tumors eventually recur and progress. Our findings suggest that HN blockade could improve the treatment of both mIDH and wtIDH gliomas.

A crucial aspect in the limited success of current therapeutic treatments lies in the highly invasive nature of GBM [33]. Our results suggest that HN induces the secretion of factors that promote GBM cell migration. The ability of GBM cells to invade also requires the capacity to modulate the extracellular matrix and remodel the tumor microenvironment. A positive correlation has been shown between the expression of MMP-2 and MMP-9, key enzymes involved in extracellular matrix degradation, and the malignancy of GBM [34]. However, we did not observe significant changes in the activity of these enzymes in the presence of HNGF6A, suggesting that the underlying mechanisms of HN on GBM cell migration may involve MMP-independent pathways. On the other hand, the interaction between tumor cells and their microenvironment is a complex process that involves bidirectional communication between tumor cells and stromal cells, such as endothelial cells, and immune cells [35,36]. The formation of new blood vessels through angiogenesis is a critical aspect of tumor expansion and invasion. Our study suggests that GBM-derived HN exerts a direct stimulatory effect on both the migration and proliferation of endothelial cells. Although more studies are needed to fully understand the mechanisms involved in the protumoral action of HN, our findings suggest a protumoral role of HN in the pathogenesis of GBM, expanding the perspective for the development of therapeutic strategies aimed at inhibiting HN function in GBM, either by the inhibition of its expression or blockade of HN receptors.

HN can be released and act as an autocrine, paracrine and endocrine messenger that interacts with membrane receptors [37]. Formylated peptide receptors (FPR) are a family of G-protein-coupled receptors. In humans, it consists of three members: FPR1, FPR2 and FPR3 [38,39]. FPR2 has been recognized as a promiscuous receptor with chemoattractant qualities that recognizes a wide variety of ligands with structural differences [39,40]. Depending on the context and the specific ligands to which FPR2 binds, this receptor can elicit responses that either promote or mitigate inflammation [41]. FPR2 overexpression has been reported in different pathologies, such as ovarian cancer [42], melanoma [43] and colon cancer [40]. In addition, FPR2 is known to exert chemotactic functions in immune cells, such as monocytes, neutrophils and dendritic cells [44]. This diverse cellular distribution suggests a multifunctional role for FPR2 in different physiological functions and pathological contexts [39]. Although the presence of FPR2 in the brain and its involvement in the inflammatory response and regulation of neuronal function have been established [22], the expression of this formylpeptide receptor in brain tumors has been described to a limited extent. Our results confirmed the expression of FPR2 in GBM cells. Moreover, we found that its expression is upregulated by chemotherapy, suggesting that this receptor could be involved in the response of GBM cells to cytotoxic injury. In fact, FPR2 blockade using a specific antagonist revealed that this receptor facilitates the survival of GBM cells under basal conditions and in response to chemotherapy, suggesting that the HN/FPR2 pathway may be important in the intrinsic resistance of GBM to cytotoxic stimuli.

Taking into account that the response of GBM to chemotherapy is very poor, several therapeutic strategies have been proposed to improve treatment results. Among them is the silencing of genes through viral vectors, which have been shown to efficiently deliver genes that block the expression of tumor-promoting genes [45,46]. In this sense, baculoviruses (BVs) represent useful tools to be used as gene therapy vectors. These vectors are relatively simple to produce and purify at high titers and have a broad cloning capacity, allowing the transfer of multiple transgenes and regulatory elements simultaneously [47,48]. Furthermore, since they are natural insect pathogens, patients do not possess pre-existing immunity against these vectors [47]. We previously reported that the local administration of BVs inhibits tumor growth in experimental rat prolactinomas [49]. Also, we have recently demonstrated that BVs efficiently transduce glioma cells and astrocytes in vitro and in vivo without causing neurotoxicity [24]. The silencing of HN transcriptional expression via BV.shHN boosted the cytotoxic response of glioma cells with heterogeneous genetic backgrounds, i.e., wtIDH and mIDH glioma cells. In addition, HN silencing impaired GBM cell migratory and proangiogenic properties, suggesting that the local administration of these vectors could constitute an interesting strategy to aid in the treatment of this aggressive tumor. Considering that HN could reach the tumor from neighboring cells, such as astrocytes [50], or even arrive to the tumor from the general circulation, local blockade of FPR2 using gene therapy vectors encoding for specific shRNA, or for the peptidic antagonist WRW4, could be useful strategies to inhibit this pathway in GBM (Figure 8).

Although treatment with TMZ has been used worldwide to treat GBM since 2005 [25,51], GBM cells exhibit strong intrinsic resistance to its cytotoxic effect. While the DNA repair enzyme MGMT has been extensively proven to be involved in the resistance of TMZ cytotoxicity [25], our findings suggest that HN could also participate in the intrinsic resistance of GBM cells to this drug. Thus, HN could constitute a therapeutic target to enhance the response of GBM cells to TMZ. In addition, considering the relative lack of efficacy of this drug in GBM, alternative chemotherapeutic drugs need to be evaluated to be used in combination or as an alternative to TMZ. Cisplatin is a very powerful antitumor drug that is gaining attention for the treatment of GBM [52,53]. However, the adverse systemic effects of cisplatin limit the efficacy of this drug. Thus, improving the sensitivity of GBM cells to lower doses of cisplatin could allow the use of this potent drug in these patients. Our findings suggest that HN protects GBM cells from both chemotherapeutic drugs. Thus, this peptide could be targeted to boost the response to combinatorial approaches. Future studies involving in vivo models and full molecular analyses are needed to elucidate the precise role of HN/FPR2 pathway in the pathogenesis of GBM and their potential as therapeutic targets.

We found that HN is expressed in the tumor as well as in normal brain tissue. Interestingly, HN expression was higher in normal brain tissue than in the tumor. This observation could be related to the fact that the biopsies deposited in the database are obtained at the time of surgery, before the patient receives chemotherapy. According to our findings, chemotherapy could upregulate HN expression. Nevertheless, HN synthesized in the non-neoplastic brain could access the tumor and facilitate the survival, chemoresistance and migration of GBM cells. Regardless of the source of HN, our findings suggest that extracellular HN-mediated cytoprotection is mediated through the activation of its membrane receptor FPR2. We found that FPR2 is upregulated in GBM biopsies and associated with a worse prognosis in GBM patients. Thus, it is possible that, regardless of the source of HN, FPR2 activation may play a relevant role in the development and progression of GBM. These observations suggest that the HN/FPR2 pathway may play a role in the pathogenesis of GBM and that it could constitute an interesting therapeutic target to improve the response of GBM cells to treatment.

## 5. Conclusions

Our findings indicate that exogenous HN exerts a robust cytoprotective effect in GBM cells, improving their viability and chemoresistance. Thus, this needs to be taken into account when designing therapies to treat chronic diseases using HN and its analogs. Endogenous HN and its FPR2 receptor could to play a role in the pathogenesis of GBM, facilitating the occurrence of the hallmark features that impair the response of GBM cells to standard treatment. Thus, blockade of HN or its receptor using BVs could be a useful strategy to improve the efficacy of chemotherapy for the treatment of GBM.

## Figures and Tables

**Figure 1 cancers-15-04061-f001:**
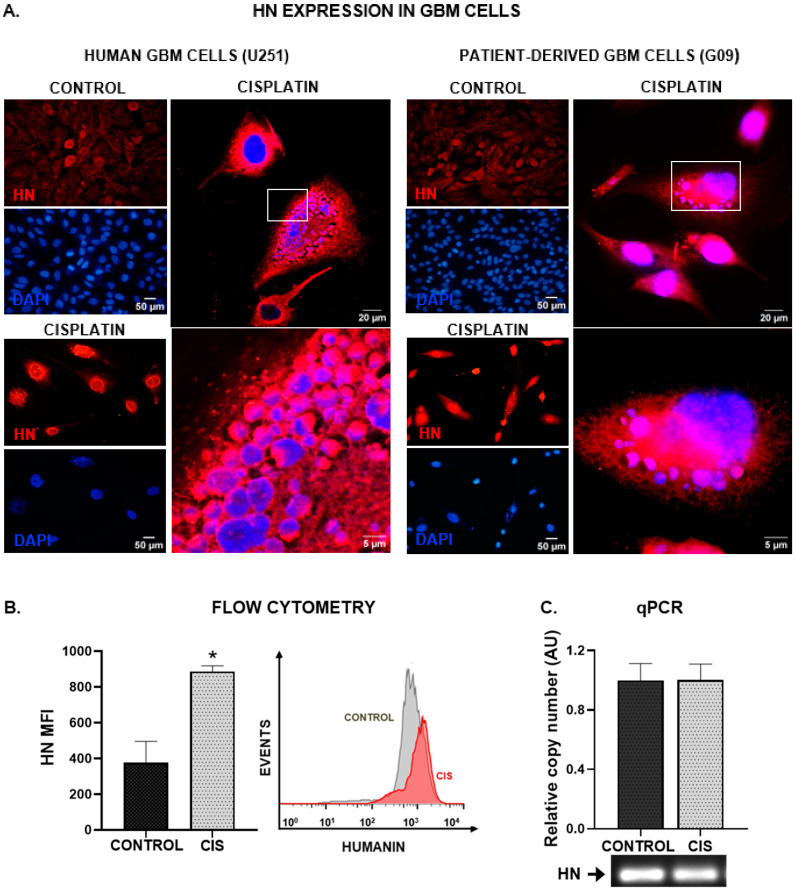
Chemotherapy upregulates HN in human GBM cells. Human U251-MG GBM cells and patient-derived GBM cell cultures (G09) were incubated with 5 µM cisplatin for 48 h. HN expression was assessed by immunofluorescence (**A**), flow cytometry (**B**) and qPCR (**C**). (**A**) Images show cells immunostained with HN antibody (red), and DAPI-stained nuclei (blue). Representative magnified images of cisplatin-treated human GBM cells were obtained by confocal microscopy. The white box indicates the area magnified in the bottom panel. (**B**) Mean fluorescence intensity (MFI) of HN staining in human U251-MG GBM cells (*n* = three replicates/condition). A representative histogram is depicted. * *p* < 0.05, Student’s *t* test. (**C**) Expression of HN mRNA as assessed by qPCR. A representative gel of qPCR products is shown.

**Figure 2 cancers-15-04061-f002:**
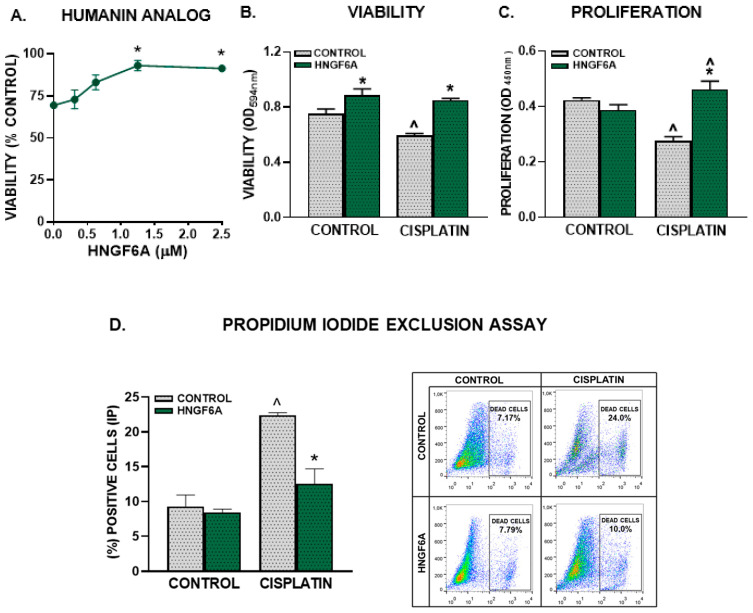
HN analogue facilitates chemoresistance in human GBM cells. (**A**–**E**) U251-MG cells were incubated with different concentrations (**A**) or 1.25 μM HNGF6A (**B**–**E**) for 2 h before cisplatin (2 μM) was added for additional 72 h (*n* = six replicates/condition). Viability was assessed by the MTT assay (**A**,**B**), proliferation was evaluated by BrdU incorporation (ELISA) (**C**) and cell death was determined by the propidium iodide exclusion method (**D**). Representative dot plots are shown for each condition. (**E**) Clonogenic capacity was evaluated 10 days after seeding the cells that were alive after cisplatin treatment (*n* = three replicates/condition). The panels on the right show representative images of the colonies formed in each experimental condition at the end of the clonogenic assay. (**F**) mIDH glioma neurospheres derived from genetically engineered mouse tumors and patient-derived biopsies cells (G01) were incubated with HNGF6A (1.25 μM) for 2 h before adding cisplatin (5 μM) for 72 h. Viability was measured by MTT assay. * *p* < 0.05 vs. respective controls without HNGF6A; ^ *p* < 0.05 vs. respective control without cisplatin, ANOVA followed by Tukey’s test.

**Figure 3 cancers-15-04061-f003:**
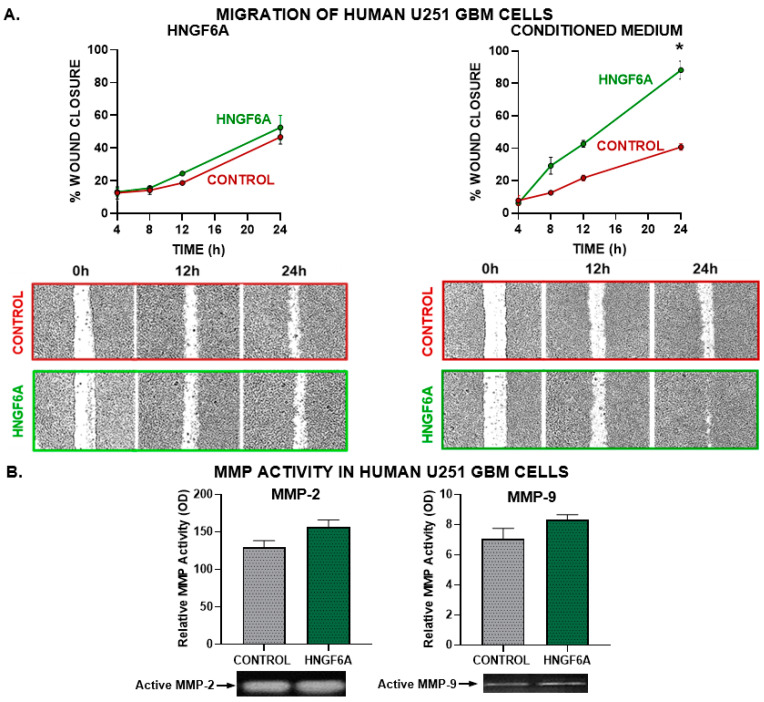
Effect of HNGF6A on GBM cell migration and angiogenic capacity. (**A**) GBM U251-MG cells were seeded to confluence and incubated with HNGF6A (1.25 μM, left panels) or with conditioned media from HNGF6A-treated cells (right panels). The monolayer was scratched and the cell-free area was measured at different time points. * *p* < 0.05 (nonlinear regression analysis). Representative images of the scratch areas are shown. (**B**) SDS-PAGE gelatin zymography of conditioned media from human GBM U251-MG cells incubated in the presence of HNGF6A (1.25 μM) for 48 h. The bands were analyzed by densitometry with the ImageJ software (Version: 1.53k) and the zymographic activity was expressed as a percentage in relation to a standard internal sample that is saturated at a density of 50%. * *p*  <  0.05 Student’s *t*-test (**C**) EA.hy926 endothelial cells were seeded to confluence and incubated directly with HNGF6A (1.25 µM) or using conditioned media from HNGF6A-treated U251-MG cells. A scratch test was performed and the cell-free area was measured at different time points. * *p* < 0.05 (nonlinear regression analysis). (**D**) EA.hy926 endothelial cells were incubated with HNGF6A (1.25 µM) or with conditioned media from HNGF6A-treated U251-MG cells for 48 h and proliferation was determined by BrdU incorporation (ELISA) * *p* < 0.05 Student’s *t*-test.

**Figure 4 cancers-15-04061-f004:**
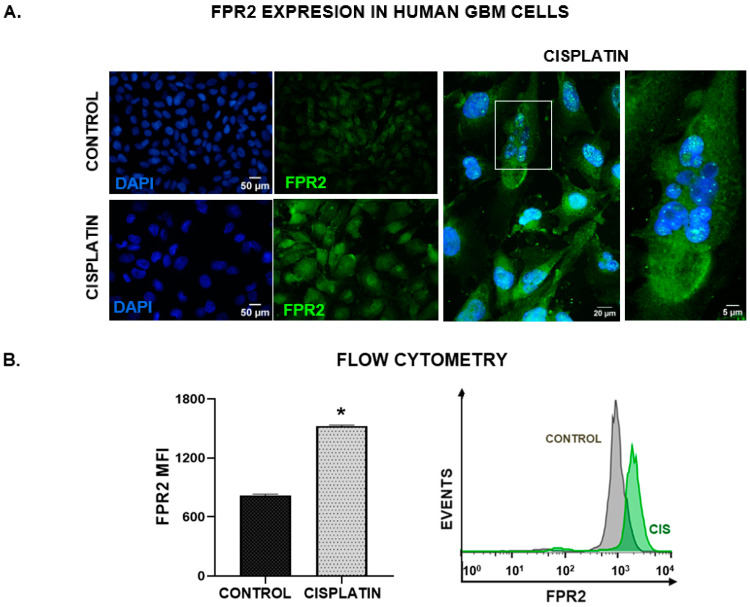
FPR2 mediates the cytoprotective effect of HN in GBM cells. Human GBM U251-MG cells were incubated with 2 µM cisplatin for 48 h. FPR2 expression was assessed by immunofluorescence (**A**) and flow cytometry (**B**). (**A**) Images show FPR2 immunostaining (green), and DAPI-stained nuclei (blue). A representative magnified image of cisplatin-treated cells using confocal microscopy is shown. The white box indicates the area magnified in the right panel (**B**) Mean fluorescence intensity (MFI) of FPR2 in human U251-MG GBM cells (*n* = three replicates/condition). * *p* < 0.05 Student’s *t* test. (**C**–**E**) U251-MG GBM cells as well as wtIDH and mIDH murine neurospheres were incubated with 10 µM WRW4 (FPR2 antagonist), in the presence of HNGF6A and cisplatin for 72 h. (*n* = six replicates/condition). Viability was determined by MTT assay (**C**,**E**) and proliferation was assessed by BrdU incorporation (ELISA, (**D**)). * *p* < 0.05 vs. respective control without HNGF6A; ^ *p* < 0.05 vs. respective control without cisplatin. + *p* < 0.05 vs. respective control without WRW4. ANOVA followed by Tukey’s test. (C: Control, H: HNGF6A).

**Figure 5 cancers-15-04061-f005:**
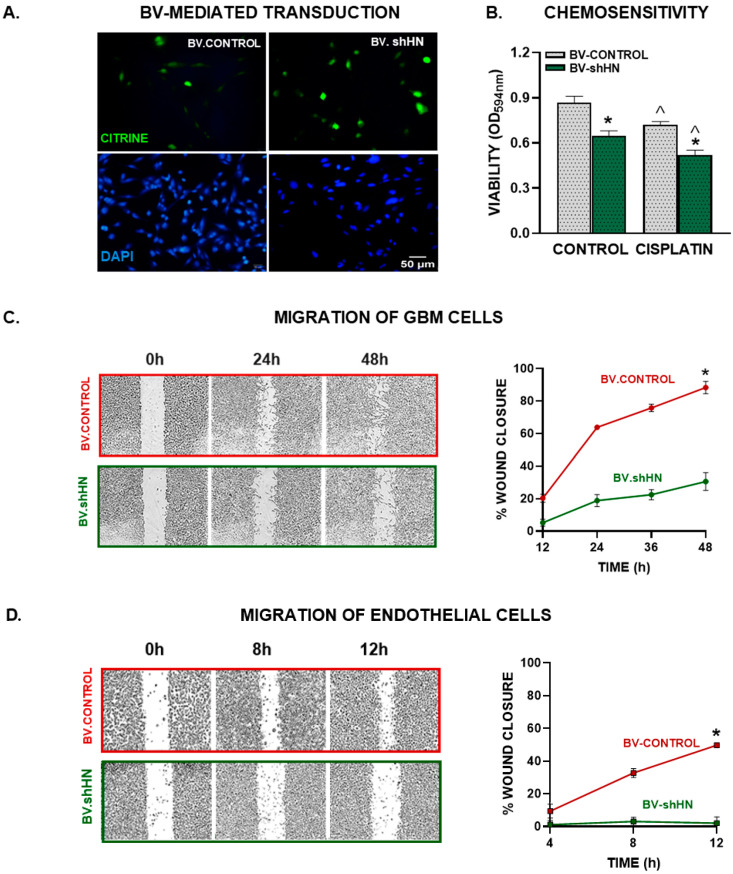
BV-mediated silencing of HN in GBM cells. Murine GBM cells (GL26) were transduced with BV.Control or BV.shHN (750 pfu/cell) for 48 h. (**A**) Expression of the reporter gene (green) was assessed using fluorescent microscopy. (**B**) Transduced cells were incubated with cisplatin (5 μM) for 72 h and viability was assessed by MTT assay. * *p* < 0.05 vs. respective BV.Control, ^ *p* < 0.05 vs. respective control without cisplatin. ANOVA followed by Tukey’s test. (**C**) Murine GBM cells (GL26) were seeded until reaching confluence, transduced with BV.shHN or BV.Control (750 pfu/cells) and migration was evaluated at different time points using the wound assay. * *p* < 0.05 vs. BV.Control (nonlinear regression analysis). (**D**) EA.hy926 endothelial cells were seeded to confluence and incubated with conditioned medium from GBM cells transduced with BV-shHN. A wound assay was performed and the cell-free area was measured at different time points. * *p* < 0.05 vs. BV.Control (nonlinear regression analysis).

**Figure 6 cancers-15-04061-f006:**
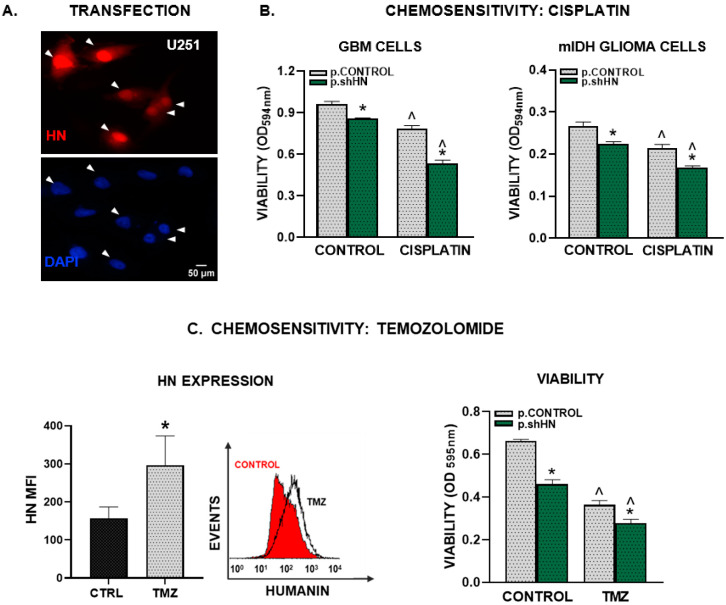
Silencing of HN and chemosensitivity in human GBM cells. Human U251-MG GBM cells as well as cells derived from mIDH glioma biopsies (G01) were transfected with a plasmid encoding an shRNA for human HN and the red fluorescent protein dtTomato, or a control plasmid not expressing the silencing sequence. (**A**) Representative images show reporter-gene-positive cells (red) and nuclei stained with DAPI (blue). Arrows indicate transfected cells. (**B**,**C**) Transfected cells were incubated with 2 μM cisplatin (**B**) or with 15 μM temozolomide, TMZ (**C**) for 72 h and viability was assessed by the MTT assay. * *p* < 0.05 vs. respective control plasmid (p.control), ^ *p* < 0.05 vs. respective control without cisplatin or TMZ. ANOVA followed by Tukey’s test. HN expression was assessed by flow cytometry in human U251-MG GBM cells that were incubated with 15 µM temozolomide for 48 h. Images show mean fluorescence intensity (MFI) (*n* = three replicates/condition). A representative histogram is depicted. * *p* < 0.05 Student’s t test (**C**).

**Figure 7 cancers-15-04061-f007:**
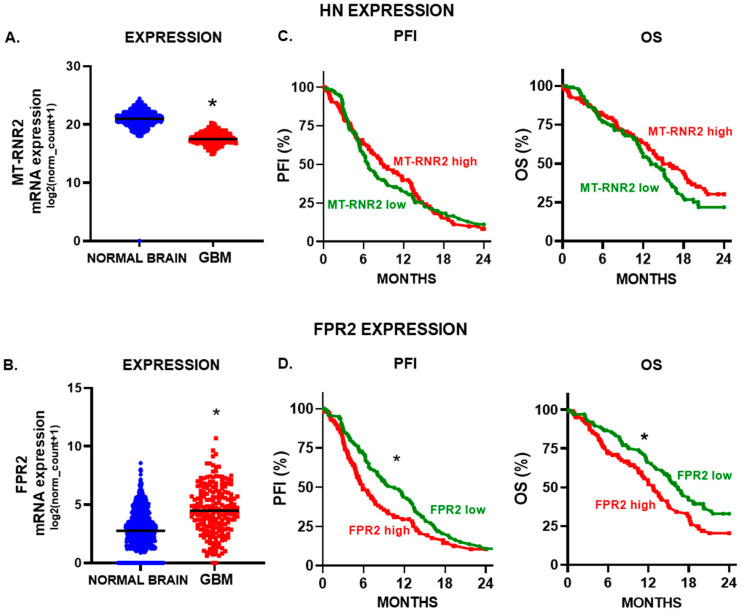
HN and FPR2 expression and prognosis of GBM patients. The mRNA expression of humanin (**A**) (MT-RNR2) and (**B**) FPR2 was evaluated using transcriptomic data of normal brain tissue (GTEx, *n* = 1141) and GBM biopsies (TCGA Pan-Cancer database, *n* = 207). *, *p* < 0.05; Mann-Whitney U test. Kaplan–Meier curves were created using UCSC Xena database and TCGA LGG-GBM cohorts. Progression-free-interval (PFI) and overall survival (OS) curves of GBM patients that were stratified according to (**C**) HN (MT-RNR2) and (**D**) FPR2 mRNA expression levels using the median of expression as a cut-off point. * *p* < 0.05, Log rank (Mantel–Cox) test.

**Figure 8 cancers-15-04061-f008:**
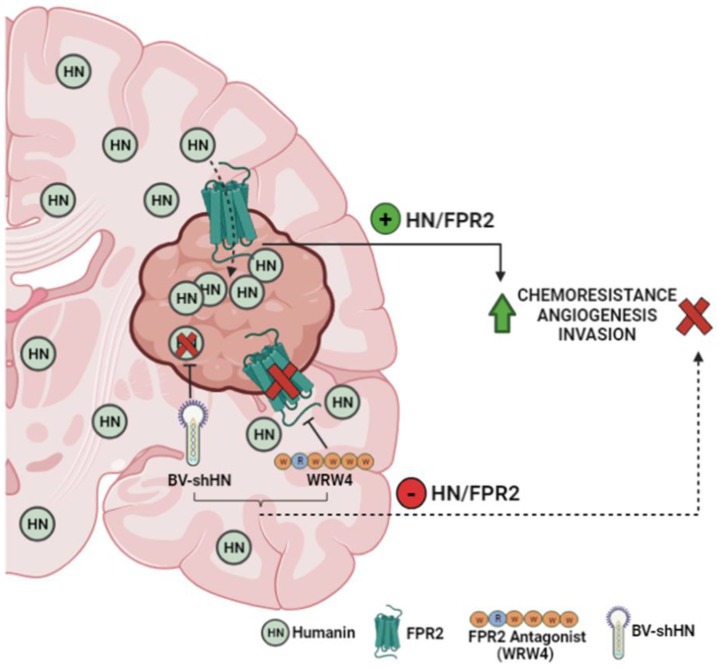
HN and FPR2 as therapeutic targets in GBM. HN, originated in brain and the tumor, can interact with the FPR2 receptor present in GBM cells, facilitating chemoresistance, angiogenesis and invasion. Transcriptional blockade of HN using BV.shHN or inhibition of FPR2 with WRW4 antagonist improves the chemosensitivity of GBM cells. Thus, the HN/FPR2 pathway could constitute a therapeutic target to improve GBM response to standard therapy, suppressing chemoresistance and reducing the invasive and angiogenic capacity of the tumor.

## Data Availability

The data can be shared up on request.

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
