# Peer review of "Mitochondrial Peptide Humanin Facilitates Chemoresistance in Glioblastoma Cells"

_cancers, 2023, doi:10.3390/cancers15164061_

Round 1

Reviewer 1 Report

This manuscript investigates the role of Humanin (HN), a cytoprotective peptide, in glioblastoma multiforme (GBM) cells and its implications for chemotherapy resistance. The study reveals that HN expression is upregulated in GBM cells following chemotherapy treatment, and exogenous HN promotes cytoprotection and chemoresistance in these cells. The authors demonstrate that HN enhances GBM cell viability, inhibits cell death and proliferation, and restores clonogenic capacity in the presence of cisplatin. Furthermore, HN influences the migratory capacity of GBM cells and endothelial cells, potentially contributing to tumor invasion and angiogenesis. The expression of the HN receptor, Formyl peptide receptor 2 (FPR2), is also observed in GBM cells and is upregulated by chemotherapy. Blockade of FPR2 using a specific antagonist attenuates the cytoprotective effects of HN in GBM cells. Silencing endogenous HN expression through viral vectors decreases GBM cell viability, sensitizes cells to chemotherapy, and inhibits migration and proangiogenic properties. The authors propose that targeting HN or its receptor could be a promising strategy to enhance the efficacy of chemotherapy in GBM treatment.

To strengthen the manuscript, several improvements can be made:

1) Lack of human clinical data: Although the study provides valuable insights using in vitro model, the lack of human clinical data limits the direct translation of the findings to the clinical setting. Including data from GBM patient samples would enhance the clinical relevance of the study and provide a better understanding of the role of HN in human GBM. This could involve evaluating HN expression levels, FPR2 activation, and the effects of HN modulation on chemoresistance, migration, and angiogenesis in these samples. The authors could also conduct the correlation studies using clinical GBM samples to assess the expression levels of HN and FPR2 (if possible with available data). Correlating HN expression levels with patient outcomes, treatment response, and overall survival would provide insights into the prognostic value and therapeutic potential of HN in GBM.

2) Investigation of HN distribution: The claim regarding the distribution of HN within or around the nucleus after chemotherapy treatment needs further investigation. Performing cell fractionation experiments and using immunoblotting to detect the change in HN distribution would provide more comprehensive insights into the intracellular trafficking of HN.

3) Duration and efficiency of HN silencing: The authors performed HN silencing experiments using p.shHN, but the duration and efficiency of HN silencing are not discussed. Providing information on the duration of silencing and the efficiency of knockdown would strengthen the interpretation of the findings.

4) HN overexpression models: Creating HN overexpression models in GBM cells or animal models would allow for a more comprehensive investigation of HN's effects on tumor behavior. Assessing changes in chemoresistance, migration, angiogenesis, and tumor growth in response to HN overexpression would provide additional evidence supporting the role of HN in GBM.

5) Combination therapy studies: Investigating the potential synergistic effects of HN-targeted therapies in combination with standard treatments or emerging therapies for GBM would be valuable. Conducting a series of combination therapy experiments using HN-targeted therapies along with chemotherapy agents to determine IC50 values or assessing their combined effects on tumor growth, survival, and therapeutic response would provide insights into the potential benefits of combining HN-targeted therapies with existing treatments.

By addressing these improvements, the manuscript would further enhance our understanding of the role of HN in GBM and provide a more comprehensive basis for potential therapeutic interventions.

Author Response

This manuscript investigates the role of Humanin (HN), a cytoprotective peptide, in glioblastoma multiforme (GBM) cells and its implications for chemotherapy resistance. The study reveals that HN expression is upregulated in GBM cells following chemotherapy treatment, and exogenous HN promotes cytoprotection and chemoresistance in these cells. The authors demonstrate that HN enhances GBM cell viability, inhibits cell death and proliferation, and restores clonogenic capacity in the presence of cisplatin. Furthermore, HN influences the migratory capacity of GBM cells and endothelial cells, potentially contributing to tumor invasion and angiogenesis. The expression of the HN receptor, Formyl peptide receptor 2 (FPR2), is also observed in GBM cells and is upregulated by chemotherapy. Blockade of FPR2 using a specific antagonist attenuates the cytoprotective effects of HN in GBM cells. Silencing endogenous HN expression through viral vectors decreases GBM cell viability, sensitizes cells to chemotherapy, and inhibits migration and proangiogenic properties. The authors propose that targeting HN or its receptor could be a promising strategy to enhance the efficacy of chemotherapy in GBM treatment.

We thank this Reviewer for the thorough revision of our paper and for the constructive comments she/he have made. We answer all her/his suggestions as follows:

To strengthen the manuscript, several improvements can be made:

1) Lack of human clinical data: Although the study provides valuable insights using in vitro model, the lack of human clinical data limits the direct translation of the findings to the clinical setting. Including data from GBM patient samples would enhance the clinical relevance of the study and provide a better understanding of the role of HN in human GBM. This could involve evaluating HN expression levels, FPR2 activation, and the effects of HN modulation on chemoresistance, migration, and angiogenesis in these samples. The authors could also conduct the correlation studies using clinical GBM samples to assess the expression levels of HN and FPR2 (if possible with available data). Correlating HN expression levels with patient outcomes, treatment response, and overall survival would provide insights into the prognostic value and therapeutic potential of HN in GBM.

We agree with this Reviewer that clinical data could provide insights into the prognostic value and therapeutic potential of HN in GBM. We have now performed a meta-analysis of transcriptomic data from GBM biopsies and non-neoplastic brain samples deposited at TCGA and GTEX, respectively. We found that HN is expressed in the tumor and in the normal brain. Interestingly, HN expression is higher in the normal brain than in the tumor. This observation could be related to the fact that the biopsies analyzed are obtained at the time of surgery, before the patient receives chemotherapy, a treatment that according to our findings could upregulate HN expression. Nevertheless, HN synthesized in the non-neoplastic brain could facilitate the survival, chemoresistance and migration of GBM cells. Regardless the source of HN, our findings suggest that extracellular HN-mediated cytoprotection is mediated through the activation of it membrane receptor FPR2. Thus, we assessed the expression of this receptor in normal and neoplastic brain (Figure 7). We found that FPR2 is upregulated in GBM when compared to non-neoplastic brain tissue. When we stratified GBM patients according to the expression of HN and FPR2, we found that while patients with high or low tumoral expression of HN exhibited similar survival curves, high tumoral expression of FPR2 was associated with worse survival. These observations suggest that the HN/FPR2 pathway may play a role on the pathogenesis of GBM and that it could constitute interesting therapeutic target to improve the response of GBM cells to treatment. We have added this information in the Material and Methods, Results and Discussion sections (Pages 5,16-17,19). We have also included a scheme summarizing our findings and conclusions (Figure 8).

2) Investigation of HN distribution: The claim regarding the distribution of HN within or around the nucleus after chemotherapy treatment needs further investigation. Performing cell fractionation experiments and using immunoblotting to detect the change in HN distribution would provide more comprehensive insights into the intracellular trafficking of HN.

We agree with this Reviewer that a more exhaustive evaluation is required to understand the intracellular trafficking of HN, and have now corrected the sentence regarding the cellular localization of HN (Page 6), as this evaluation exceeds the aims of our work. Our aims were: (i) to evaluate whether exogenous HN analogs, which were proposed as therapeutic tools to treat chronic diseases such as neurodegenerative disorders, could affect the pathophysiology of GBM cells, and (ii) whether this antiapoptotic mitochondrial peptide could constitute a therapeutic target to improve the response of GBM to chemotherapy. Thus, we evaluated whether HN and FPR2 were actually expressed in GBM cells, and whether their expression could be affected by chemotherapy. We found that that GBM cells upregulate both, HN and FPR2, and that their blockade sensitizes these cells to chemotherapy.

3) Duration and efficiency of HN silencing: The authors performed HN silencing experiments using p.shHN, but the duration and efficiency of HN silencing are not discussed. Providing information on the duration of silencing and the efficiency of knockdown would strengthen the interpretation of the findings.

We have previously recently demonstrated that BV are excellent tools for transgene delivery to GBM cells in vitro and in vivo (Garcia Fallit, Pidre et al. 2023). Here, we developed a BV vector that encodes for a shRNA to silence HN and show the high transduction efficiency that these vectors display in GBM cells. The extent of HN silencing using BV-vectors has been previously reported by we and others (Marvaldi, Martin et al. 2021). HN BV-mediated silencing has been shown to inhibit the expression of this peptide and to boost apoptosis in cells of the pituitary gland and ovarian granulosa 24 and 48 h after transduction (Gottardo, Pidre et al. 2018, Marvaldi, Martin et al. 2021). In addition, we have previously shown HN blockade at the protein level in murine breast cancer cells using this shRNA sequence (Moreno Ayala, Gottardo et al. 2020). Our study shows that silencing endogenous humanin using BV.shHN elicits significant effects in GBM cells. It enhances chemosensitivity, reduces invasion and angiogenic potential in GBM. This approach represents a promising strategy to improve the response to standard therapy and develop more effective targeted treatments against GBM.

4) HN overexpression models: Creating HN overexpression models in GBM cells or animal models would allow for a more comprehensive investigation of HN's effects on tumor behavior. Assessing changes in chemoresistance, migration, angiogenesis, and tumor growth in response to HN overexpression would provide additional evidence supporting the role of HN in GBM.

This is a very interesting point. However, it goes beyond our aim. As we have mentioned above, we are in seek of therapeutic alternatives to sensitize GBM cells to chemotherapy. In addition, overexpression of HN would constitute an artifact that may not mimic what happens in GBM cells, as it would result in very high and unregulated levels of HN. On the other hand, HN can access the tumor from neighboring or even distant tissues, and enter GBM cells or interact with membrane receptors. Our findings suggest that FPR2 plays a central role on the cytoprotective effects of endogenous and exogenous HN.

5) Combination therapy studies: Investigating the potential synergistic effects of HN-targeted therapies in combination with standard treatments or emerging therapies for GBM would be valuable. Conducting a series of combination therapy experiments using HN-targeted therapies along with chemotherapy agents to determine IC50 values or assessing their combined effects on tumor growth, survival, and therapeutic response would provide insights into the potential benefits of combining HN-targeted therapies with existing treatments.

We thank this Reviewer for this suggestion. Due to regulatory restrictions, we cannot conduct these experiments at this point, but we will evaluate the efficacy of HN and FPR2 blockade in combination with chemotherapy in relevant models of GBM in the near future.

By addressing these improvements, the manuscript would further enhance our understanding of the role of HN in GBM and provide a more comprehensive basis for potential therapeutic interventions.

We have now included all the suggestions made by this Reviewer in the revised version of our manuscript.

REFERENCES

Garcia Fallit, M., M. L. Pidre, A. S. Asad, J. A. P. Agudelo, M. B. Vera, A. J. N. Candia, S. B. Sagripanti, M. P. Kuper, L. C. A. Morales, A. Marchesini, N. Gonzalez, C. M. Caruso, V. Romanowski, A. Seilicovich, G. A. Videla-Richardson, F. A. Zanetti and M. Candolfi (2023). "Evaluation of Baculoviruses as Gene Therapy Vectors for Brain Cancer." Viruses 15(3): 608.

Gottardo, M. F., M. L. Pidre, C. Zuccato, A. S. Asad, M. Imsen, G. Jaita, M. Candolfi, V. Romanowski and A. Seilicovich (2018). "Baculovirus-based gene silencing of Humanin for the treatment of pituitary tumors." Apoptosis 23(2): 143-151.

Marvaldi, C., D. Martin, J. G. Conte, M. F. Gottardo, M. L. Pidre, M. Imsen, M. Irizarri, S. L. Manuel, F. E. Duncan, V. Romanowski, A. Seilicovich and G. Jaita (2021). "Mitochondrial humanin peptide acts as a cytoprotective factor in granulosa cell survival." Reproduction 161(5): 581-591.

Moreno Ayala, M. A., M. F. Gottardo, C. F. Zuccato, M. L. Pidre, A. J. Nicola Candia, A. S. Asad, M. Imsen, V. Romanowski, A. Creton, M. Isla Larrain, A. Seilicovich and M. Candolfi (2020). "Humanin Promotes Tumor Progression in Experimental Triple Negative Breast Cancer." Sci Rep 10(1): 8542.

Reviewer 2 Report

A fine paper, in need of some minor modifications, in the materials and methods section. You used a large variety of materials in achievements of your results. Not all of these are properly indicated in the material and method section; for instance FBS , Culture media  - DMEM, DMEM-F12,catalogue No) since many are available and several variants also usually from the same provider). Please revise the section and put the required information. 

Author Response

A fine paper, in need of some minor modifications, in the materials and methods section. You used a large variety of materials in achievements of your results. Not all of these are properly indicated in the material and method section; for instance, FBS, Culture media - DMEM, DMEM-F12, catalogue No) since many are available and several variants also usually from the same provider). Please

revise the section and put the required information.

We thank this Reviewer for the careful revision of our manuscript. We are thrilled to hear that she/he finds it as a “fine paper”. We apologize for the lack of precision in the material and methods section and have now provided the information missing.